# Crosslinked Fluorinated Poly(arylene ether)s with POSS: Synthesis and Conversion to High-Performance Polymers

**DOI:** 10.3390/polym13203489

**Published:** 2021-10-11

**Authors:** Yashu He, Jingting Wang, Igor S. Sirotin, Vyacheslav V. Kireev, Jianxin Mu

**Affiliations:** 1Key Laboratory of High Performance Plastics, Ministry of Education, National & Local Joint Engineering Laboratory for Synthesis Technology of High Performance Polymer, College of Chemistry, Jilin University, 2699 Qianjin Street, Changchun 130012, China; yshe20@mails.jlu.edu.cn (Y.H.); wangjt17@mails.jlu.edu.cn (J.W.); 2Faculty of Petrochemistry and Polymer Materials, Mendeleev University of Chemical Technology of Russia, Miusskaya sq. 9, 125047 Moscow, Russia; isirotin@muctr.ru (I.S.S.); kireev@muctr.ru (V.V.K.)

**Keywords:** crosslinked poly(arylene ether)s, POSS, high-performance polymers

## Abstract

This study reports on a series of crosslinked poly(arylene ether)s with POSS in the main chain. The fluorinated and terminated poly(arylene ether)s were first synthesized by the nucleophilic reaction of diphenol POSS and decafluorodiphenyl monomers, including decafluorobiphenyl, decaflurobenzophenone, and decafluorodiphenyl sulfone. They were then reacted with 3-hydroxyphenyl acetylene to produce phenylacetylene-terminated poly(arylene ether)s. The polymers were of excellent processability. When heated to a high temperature, the polymers converted into a crosslinked network, exhibiting a low range of dielectric constant from 2.17 to 2.58 at 1 HMz, strong resistance against chemical solutions, low dielectric losses, and good thermal and hydrophobic properties.

## 1. Introduction

Fluoropolymers have excellent thermal resistance, optical and electronic properties, thermal stability, hydrophobicity, and solvent resistance. Therefore, they have been widely used in the aerospace, construction, petrochemical, and automobile industries [1,2,3,4]. However, due to the high crystallinity of perfluorinated polymers (such as polytetrafluoroethylene), their workability is poor. Therefore, many modified, partially fluorinated polymers, such as fluorinated poly(arylene ether sulfone), poly(arylene ether)s, polyimide, and other materials possessing good workability and dielectric properties, have been studied. These properties allow for materials to be applied to the semiconductors, integrated circuits, and optoelectronics fields [5,6,7]. Compared with other fluorinated materials whose fluorine groups are nonsymmetric substituents, poly(arylene ether)s with fluorine groups as symmetric substituents tend to achieve a lower dielectric constant, due to the symmetric substitution of fluorine groups reducing the dipole moment and the dielectric constant of the materials. To obtain better, low-dielectric materials, adding nanoparticles in the polymer is a common and efficient method.

POSS is a nanomaterial with a core-shell structure containing an organic–inorganic hybrid, from which a variety of organic groups can attach to the external silicon atom. POSS has aroused considerable interest among researchers as a “self-healing” material [8,9], and as a template for preparing nanomaterials [10], catalysts [11,12], and polymers of various shapes [13,14]. Generally, the addition of POSS can effectively improve the hydrophobicity, thermal stability, mechanical, and dielectric properties of polymer materials [15,16,17]. 2OH-DDSQ is a bi-functional double-layer-structured POSS, usually used as a filler to synthesize linear polymers containing POSS in the main chain [18,19,20].

For novel dielectric materials, some properties such as dielectric properties, solubility, hydrophobicity, and adhesion to some inorganic materials are crucial [21]. Among many requirements, thermal performance is the most difficult performance of low-dielectric materials. In the processing of low-dielectric materials, the polymer material used as the interlayer dielectric is resistant to temperatures exceeding 300 °C. Crosslinked polymers have outstanding thermal stability, mechanical properties, and chemical resistance, and the dielectric constant of the polymer can be further lowered due to the chemical structure formed in the reaction [22,23]. In addition, crosslinked polymers have excellent processability. These polymers possess good solubility in organic solvents before curing, and as temperature increases, a crosslinked network produced by a cycloaddition reaction has a good solvent resistance.

In this study, the fluorinated and terminated poly(arylene ether)s with POSS in the main chain (DDSQ-DFPXs) were synthesized and then reacted with 3-hydroxyphenyl acetylene (PAL) to obtain a series of phenylacetylene-terminated poly(arylene ether)s with POSS in the main chain (DDSQ-DFPXs-PAL), which can be cured under an appropriate temperature. Additionally, a thermogravimetric analyzer, differential scanning calorimeter, Fourier transform infrared spectrometer, X-ray diffractometer, precision impedance analyzer, and contact goniometer, as well as other techniques were employed to characterize the phenylacetylene terminated poly(arylene ether)s.

## 2. Materials and Methods

### 2.1. Materials

The diphenol POSS (2OH-DDSQ) was prepared by our laboratory [24]. Decafluorobiphenyl (DFP) and decaflurobenzophenone (DFPK) were provided by Sigma-Aldrich Chemical Co. Ltd., (Changchun, China) and used after recrystallization in solution. Decafluorodiphenyl sulfone (DFPS) was synthesized in our laboratory, and all three of the above decafluorodiphenyl monomers were called DFPX. 3-hydroxyphenyl acetylene (PAL) was purchased from Sigma-Aldrich Chemical Co. Ltd., Changchun, China, and used without being further purified. N-methyl-2-pyrrolidone (NMP); N, N-dimethylacetamide (DMAC); N,N-dimethylformamide (DMF), dimethylsulfoxide (DMSO), tetrahydrofuran (THF), and chloroform were supplied by Aladdin Biochemical Technology Co. Ltd., Changchun, China.

### 2.2. Method 

The thermal stability analysis of membranes was carried out using a thermogravimetric analyzer (Pryis 1 TGA, Perkin Elmer, Waltham, MA, USA) under an air atmosphere. The specific operation steps were as follows: First, we heated the samples from 25 to 100 °C at a rate of 10 °C/min and were kept isothermal for 20 min. Then, the samples were heated to 800 °C at the same rate, and the temperatures at 5% and 10% mass losses of each sample were recorded. A differential scanning calorimeter (DSC 821e, Mettler Toledo, Greifensee, Switzerland) was used from 40 to 250 °C at a rate of 10 °C/min under a nitrogen atmosphere. The infrared spectra were collected using a Fourier transform infrared spectrometer (Bruker V70, BRUKER, Berlin, Germany) at 25 °C. ^1^H NMR and ^19^F NMR spectra were obtained using a spectrometer (Bruker-Avance III spectrometer 400MHz, BRUKER, Berlin, Germany). An X-ray diffractometer (Empyrean, PANalytical B.V., Almelo, The Netherlands) with CuKa radiation as the X-ray source was used to study the crystallization behavior of the polymers. Dielectric constants were determined by a precision impedance analyzer (Hewlett-Packard 4285A, Palo Alto, CA, USA) at frequency scope varied from 10^3^ to 10^6^ Hz. The films were filled with silver paste at the diameter of 5.5 mm, and then stoved. Contact angle (CA) measurements were prepared using a contact goniometer (POWEREACH/JC2002C2 CA meter, Shanghai, China) at room temperature.

### 2.3. Polymer Synthesis

#### 2.3.1. Synthesis of DDSQ-DFPXs-PAL

As shown in Figure 1, the fluorinated and terminated DDSQ-DFP was synthesized by the reaction of the corresponding decafluorodiphenyl monomers and 2OH-DDSQ. A three-necked flask was added to 2OH-DDSQ (1.4822 g), anhydrous K_2_CO_3_ (0.1658 g), DMAC (9 mL), and toluene (4.5 mL). After refluxing at 140 °C for 3 h and removing the toluene, the reaction mixture was cooled to 30 °C, and DFP (0.3675 g) was then added. The temperature was kept for 2 h. Finally, the reactants were poured into 50 mL of ethanol and washed several times with deionized water and ethanol solution. The samples were dried at 120 °C for 24 h for the next reaction.

The DDSQ-DFPXS-PAL was synthesized from fluorinated and terminated DDSQ-DFP and PAL. Taking DDSQ-DFP-PAL as an example, fluorinated and terminated DDSQ-DFP (2 g), CsF (0.06 g), PAL (0.236 g), and DMAC (9 mL) were added into a three-necked flask equipped with a mechanical stirrer, a nitrogen inlet, and a condenser. The temperature was raised to 120 °C and maintained for 8 h. Then, the reacted solution was poured into a 50 mL ethanol solution and washed several times with deionized water and ethanol solution. Finally, DDSQ-DFP-PAL was obtained after drying.

#### 2.3.2. Film Preparation

The crosslinked films were obtained from the thermopolymerization of DDSQ-DFPXs-PAL at high temperatures. The above-mentioned polymers were dissolved in DMAC, and then spin-coated onto a 2 × 5 cm^2^ smooth, Teflon plate. After reaching a temperature of 120 °C for 12 h, the films were heated at 270 °C for 1 h to obtain transparent, crosslinked films.

## 3. Results and Discussion

### 3.1. Polymer Synthesis and Characterization

The fluorinated and terminated DDSQ-DFPXs have a general formula, as shown in Figure 1. The X linkages of the monomers are none for DFP, ketone for decafluorobenzone (DFPK), and sulfone for decafluorodiphenyl sulfone (DFPS). The polymerization of DFPX and 2OH-DDSQ was carried out in a two-step nucleophilic aromatic substitution process. In this approach, 2OH-DDSQ was first converted to phenoxide and then further reacted with perfluoro-monomers at a low temperature. The ^1^H NMR and ^19^F NMR spectra were measured in a spectrometer (Bruker-Avance III spectrometer 400 MHz, BRUKER, Berlin, Germany) using deuterated dimethyl sulfoxide as the solvent. The ^19^F spectra are shown in Figure 2, where the spectra contain clear peaks. The corresponding positions of each peak are shown in the figure: a and b, respectively, represent the meta-position and ortho-position of fluorine elements on the polymer skeleton; a’, b’, and c represent the meta-position, ortho-position, and para-position of the fluorine elements of the fluorinated and terminated DDSQ-DFPXs, respectively; the relative integral intensities c:a’:b’ = 1:2:2; and the ratios of the relative integral intensities a:a’ and b:b’ are both equal to 2:1, where the appearance of peaks a’, b’, and c index the correctness of the polymer structure.

The fluorinated and terminated DDSQ-DFPXs were further reacted with PLA to obtain DDSQ-DFPXs-PAL, which can be self-crosslinked. The materials were characterized by FTIR, and the results are shown in Figure 3. The infrared spectra of the materials show that the characteristic absorption peaks of Si–C are at 852 and 1253 cm^−1^, while 1071 and 1133 cm^−1^ represent the asymmetric absorption peaks of Si–O–Si. The presence of the above absorption peaks indicates the presence of the POSS structure. In addition, 3301 cm^−1^ is the stretching vibration peak of C–H on the phenylacetylene terminal alkyne, and the stretching vibration peak of the C≡C group at 2120 cm^−1^ indicates the alkynyl structure of the end group of the materials. The ^19^F spectra of the polymers are shown in Figure 4. Compared with Figure 2, peaks a’ and b’ still exist while peak c disappears due to the reaction between the terminated fluorine atom and PLA. The polymers were also identified by ^1^H NMR spectroscopy. As observed from Figure 5, 0.02, 0.54–2.58, and 3.63 ppm proton peaks are assigned to protons on the methyl, methylene, and methoxyl groups, respectively. The peak between 6.5 and 8.3 ppm represents the proton hydrogen on the benzene ring. The peak appearing at 3.56 ppm represents the atom of terminated alkynyl, indicating that the terminated alkynyl was introduced in the main chain.

### 3.2. Thermo-Crosslinking Reaction

Upon heating, C≡C groups tend to form a crosslinked network via a cycloaddition reaction. The cycloaddition reaction processed was monitored in DSC, and the results are shown in Figure 6a–c. As shown in Figure 6a, DDSQ-DFP-PLA exhibits a glass transition temperature (Tg) at 138 °C and an exothermic peak from 225 to 325 °C, which indicates the wide process window of crosslinking. From the secondary scan curve, a clear thermal glass transition occurred, showing that the first crosslinking was relatively complete. The other two curves follow the same tendency. The results are shown in Table 1. It can be seen that the Tg of the crosslinked polymers are higher than that before, providing a higher use temperature and a broader range ability.

In many cases, thermo-crosslinking can effectively improve the heat resistance of polymers. In our work, the thermostabilities of crosslinked DDSQ-DFPXs-PAL were investigated by TGA and the results are shown in Figure 7 and Table 2. As observed from the figure, the temperatures at 5% mass loss (Td_5_) of the crosslinked DDSQ-DFPXs-PAL in air atmosphere are in the range of 354–412 °C and the 10% mass loss (Td_10_) temperatures are in the range of 380–434 °C, indicating that polymers have better thermal stability. The degradation process of the crosslinked DDSQ-DFPXs-PAL is mainly divided into two steps. The first step originates from the degradation of the active alkyl chain, and the second step originates from the degradation of the skeleton, such as the aromatic rings and POSS cages. The residual rates of the polymers are higher than 30%, due to the SiO_2_ structure of the DDSQ part during the thermal decomposition process. Compared with DDSQ-DFPXs, DDSQ-DFPXs-PAL materials have a higher glass transition temperature.

### 3.3. Crystallinities and Solubility

The crystallinity of the crosslinked polymers was characterized by WAXRD, as shown in Figure 8. All the WAXRD curves contain two broad peaks at about 7.1–7.3° and 18.5–19.0°, which indicates that polymers are amorphous in nature [25,26,27]. This is due to the large volume of nano-sized POSS, which is not conducive to the regularity of the molecule chains, and which also inhibits close packing of the polymer chains. Therefore, the materials cannot exhibit crystallinity.

The ideal optoelectronic materials are soluble in the process and facilitate the film formation of the solution. After processing, however, the polymers need good solvent resistance, i.e., insolubility. The solubility of the crosslinked polymers is shown in Table 3. All materials exhibited good solvent resistance after thermal crosslinking and were almost insoluble in any organic agent. In the presence of a large amount of organic solvents, the solubility was also extremely small, at only 0.1 g/mL.

DDSQ-DFPXs can be dissolved in most common organic solvents (e.g., DMF, DMSO, NMP, or DMAC) at room temperature, while the crosslinked DDSQ-DFPXs-PAL materials are insoluble in most organic solvents, even upon heating.

### 3.4. Surface Wettability

Since water molecules are highly polar molecules, the hydrophobic properties of the material have an important influence on the dielectric properties of the material. The hydrophobicity of the material ensures the stability of the material’s dielectric constant. The surface wettability of the polymers after crosslinking was measured by the water contact angle (WCA), and the test results are shown in Figure 9 and Table 2. As the contact angles of crosslinked polymers were all higher than 90°, the polymers can thus be regarded as hydrophobic materials. Among them, the water contact angle of DDSQ-DFP-PAL was up to 103°, which is close to the water contact angle of Teflon-type materials. Due to the reduced polarity of the three materials, the water contact angle of the polymer increased from 96° for DFPS to 103° for DFP.

### 3.5. Dielectric Constant

The dielectric constant and dielectric loss of crosslinked DDSQ-DFPXs-PAL were measured according to the standard capacitance method, and the results are illustrated in Figure 10. As shown in Figure 10a, when the frequency is 1 MHz, the dielectric constant values are in the range of 2.17–2.58, where crosslinked DDSQ-DFPK-PAL exhibits the lowest dielectric constant, 2.17. Compared with the other two materials, DDSQ-DFPK-PAL has the highest symmetry, the rigid biphenyl structure makes the polymer molecular chains neatly arranged, and the free volume is relatively small, leading to an increase in the number of molecules per cubic meter. Therefore, the dielectric constant of DDSQ-DFP-PAL is larger than that of DDSQ-DFPK-PAL. On the other hand, since the molecular chain of DDSQ-DFPS-PAL is more polar than DDSQ-DFPK-PAL, the dielectric constant of DDSQ-DFPS-PAL is also larger than that of DDSQ-DFPK-PAL.

As observed from Figure 10 (b), the dielectric loss factors of crosslinked DDSQ-DFPXs-PAL at 1 MHz are in the range of 1.1 × 10^−2^–2.3 × 10^−2^, which are at a low scale. Since the crosslinked DDSQ-DFP-PAL has the highest molecular symmetry compared to the other two materials, its dielectric loss value is the lowest, with a value of 1.1 × 10^−2^.

## 4. Conclusions

In this study, a series of fluorinated and terminated DDSQ-DFPXs were synthesized via a two-step nucleophilic aromatic substitution procedure. The polymers were then further reacted with PLA to obtain DDSQ-DFPXs-PAL, and formed a crosslinked network via a cycloaddition reaction under specific temperature to acquire crosslinked DDSQ-DFPXs-PAL. All polymers exhibit a low dielectric constant, low dielectric loss, and hydrophobic properties. Among them, crosslinked DDSQ-DFPK-PAL demonstrates the lowest dielectric constant (2.17 at 1 MHz), while crosslinked DDSQ-DFP-PAL shows the lowest dielectric loss factor at 1.1 × 10^−2^ at 1 MHz. The best hydrophobicity is exhibited by crosslinked DDSQ-DFP-PAL whose contact angle is 103°, which is close to that of Teflon-type materials. All crosslinked DDSQ-DFPXs-PAL show good solvent resistance and are almost insoluble in any organic agent.

## Figures and Tables

**Figure 1 polymers-13-03489-f001:**
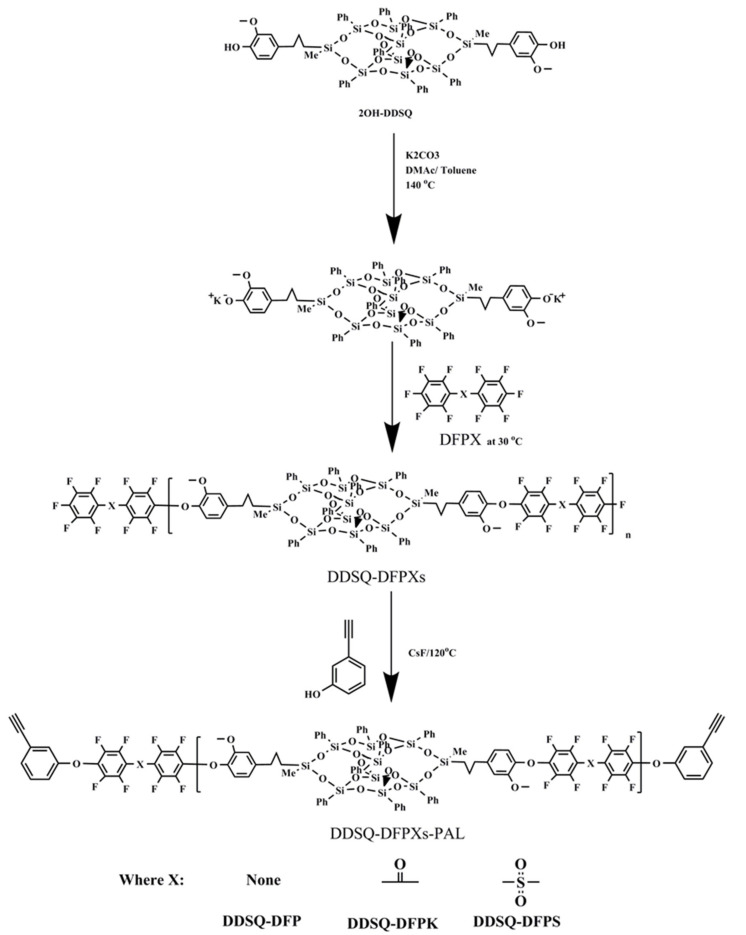
The schematic of the synthesis of DDSQ-DFPXs-PAL.

**Figure 2 polymers-13-03489-f002:**
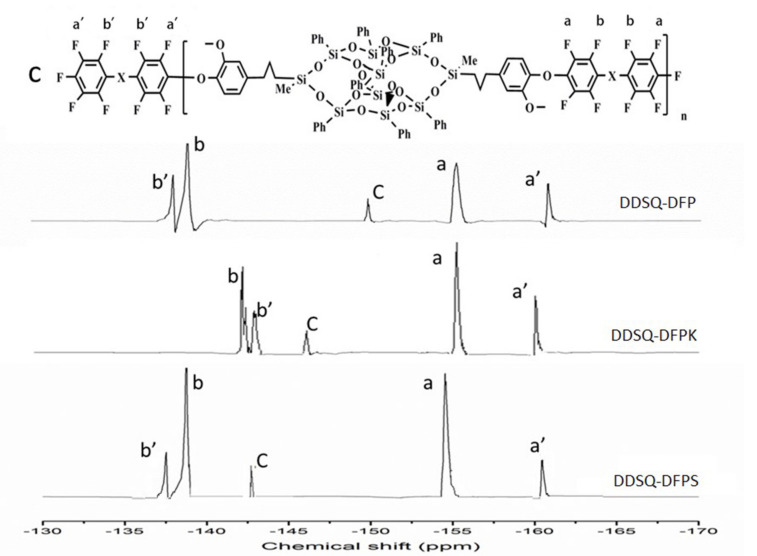
^19^F NMR spectra of fluorinated and terminated DDSQ-DFPXs in DMSO-d_6_.

**Figure 3 polymers-13-03489-f003:**
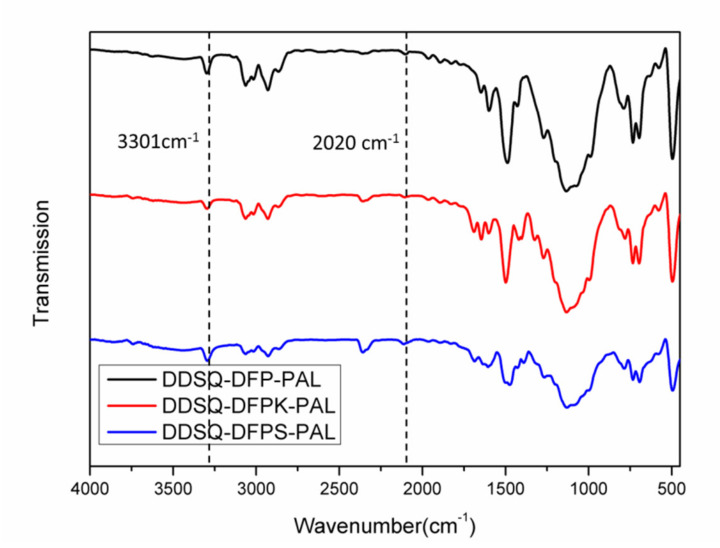
FTIR spectrum of phenylacetylene-terminated DDSQ-DFPXs-PAL.

**Figure 4 polymers-13-03489-f004:**
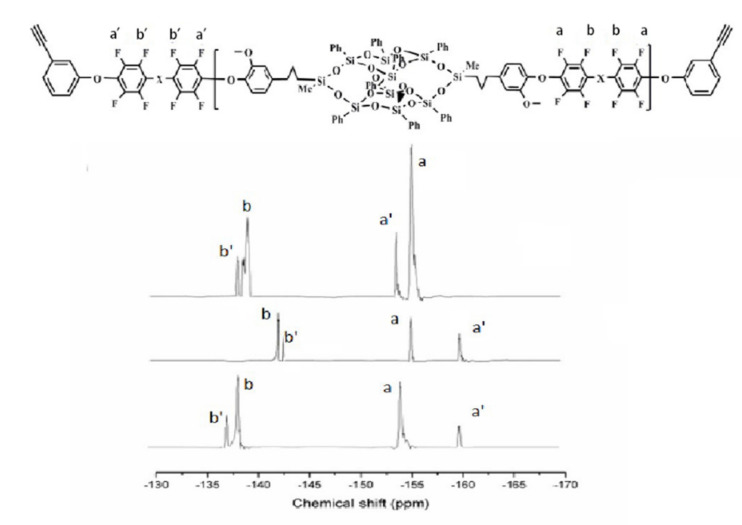
^19^F NMR spectra of phenylacetylene-terminated DDSQ-DFPXs-PAL in DMSO-d_6_.

**Figure 5 polymers-13-03489-f005:**
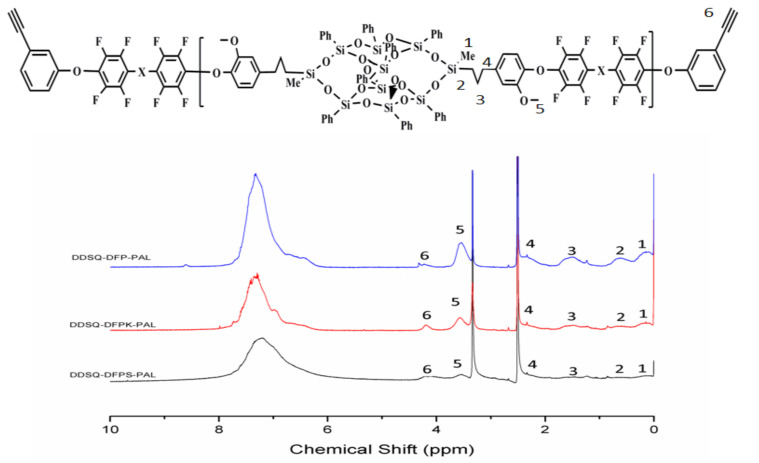
^1^H NMR spectra of phenylacetylene-terminated DDSQ-DFPXs-PAL in DMSO-d_6_.

**Figure 6 polymers-13-03489-f006:**
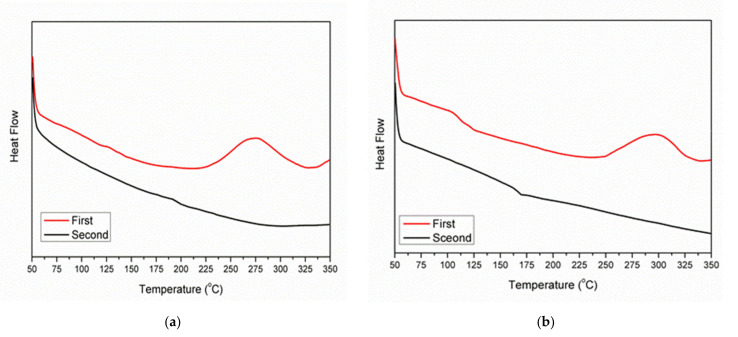
(**a**) DSC curves of the DDSQ-DFP-PAL; (**b**) DSC curves of the DDSQ-DFPK-PAL; (**c**) DSC curves of the DDSQ-DFPS-PAL.

**Figure 7 polymers-13-03489-f007:**
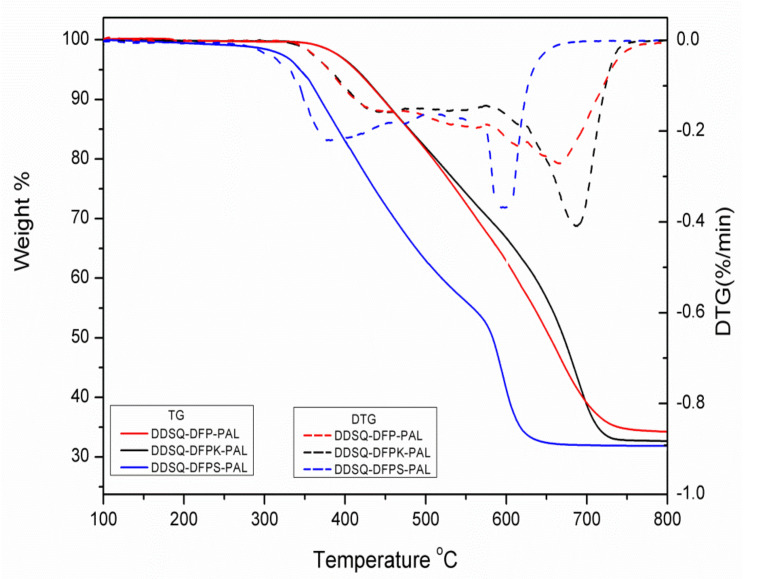
TGA curves of DDSQ-DFPKs-PAL after curing.

**Figure 8 polymers-13-03489-f008:**
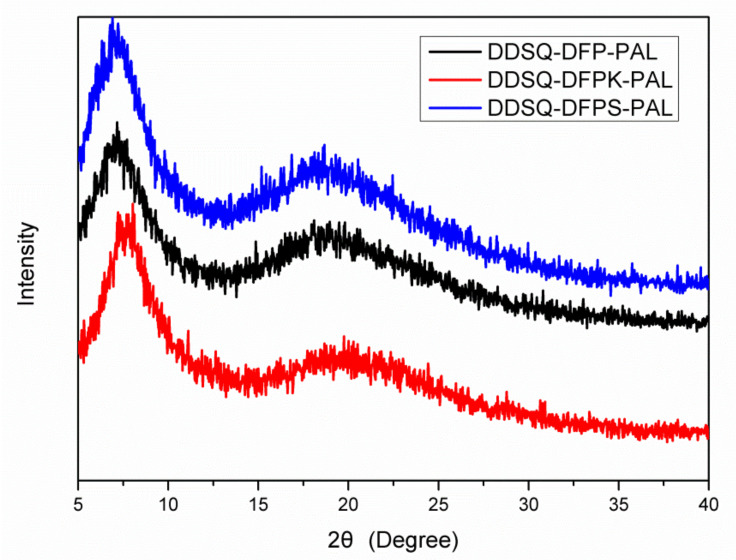
WAXRD patterns of DDSQ-DFPXs-PAL after curing.

**Figure 9 polymers-13-03489-f009:**
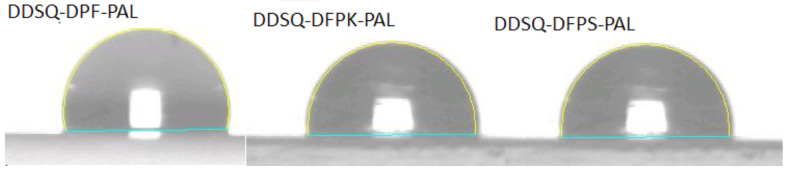
Water contact angles of DDSQ-DFPXs-PAL films after curing.

**Figure 10 polymers-13-03489-f010:**
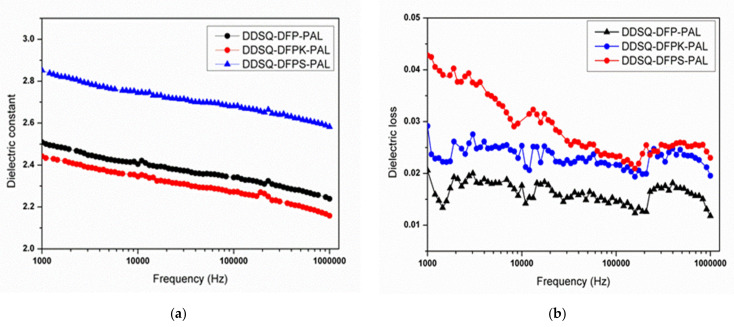
(**a**) Dielectric constant spectra of DDSQ-DFPXs-PAL films after curing; (**b**) dielectric loss spectra of DDSQ-DFPXs films after curing.

**Table 1 polymers-13-03489-t001:** DSC data of DDSQ-DFPXs-PAL.

DDSQ-DFPXs-PAL	Tg (°C before Curing)	Tg (°C after Curing)	Temperature of Curing (°C)
DDSQ-DFP-PAL	138	192	225–325
DDSQ-DFPK-PAL DDSQ-DFPS-PAL	123 145	175 187	250–325 220–330

**Table 2 polymers-13-03489-t002:** The properties of DDSQ-DFPXs-PAL after curing.

DDSQ-DFPXs-PAL (after Curing)	Td_5_ (°C)	Td_10_ (°C)	Residue (%)	Dielectric Constant	Dielectric Loss	CA (deg)
DDSQ-DFP-PAL	412	434	32.9	2.23	1.1 × 10^−2^	103
DDSQ-DFPK-PAL	402	440	31.9	2.17	1.9 × 10^−2^	99
DDSQ-DFPS-PAL	354	380	31.2	2.58	2.3 × 10^−2^	96

**Table 3 polymers-13-03489-t003:** Solubility of DDSQ-DFPXs-PAL ^a^ after curing.

DDSQ-DFPXs-PAL	DMF	DMAC	NMP	DMSO	THF	Chloroform
DDSQ-DFP-PAL	--	--	--	--	--	--
DDSQ-DFPK-PAL	--	--	--	--	--	--
DDSQ-DFPS-PAL	--	--	--	--	--	--

^a^ and --, insoluble even upon heating.

## Data Availability

The data presented in this study are openly available in [repository name e.g., FigShare] at [doi], reference number [reference number].

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
