# Peer review of "Crosslinked Fluorinated Poly(arylene ether)s with POSS: Synthesis and Conversion to High-Performance Polymers"

_polymers, 2021, doi:10.3390/polym13203489_

Round 1
Reviewer 1 Report
Mu et al. reported the synthesis of three novel short chain polymers constructed from fluorinated poly(arylene ether)s attached to an organic-inorganic hybrid core. These short chains can be crosslinked at 270 C to form transparent, hydrophobic, organic solvent resistant films. Studies of the structure of the parent chains, the process of thermo-crosslinking, and some physical properties of the film have been carried out. The main results of the study are clear presented, the paper is easy to read.
Major comment:
Please briefly review the properties of other materials made from the same organic-inorganic hybrid core. Are the new materials better than the old ones?
Minor comments:
- Fig. 2. I think that ortho-fluorines (b,b') resonate in the -155 to -160 ppm range, and metha-fluorines (a,a') resonate in the -135 to -140 ppm range. Please check.
- Fig. 2. Are the relative integral intensities c:a’:b’ = 1:2:2?
- Fig. 2. What are the ratios of the relative integral intensities a:a’ and b:b’? They provide an estimate of the average structure of the parent chains.
- Note that the crystallinity and chain dynamics in the crosslinked films can be studied using solid-state NMR (see 10.1007/s10971-018-4774-z, 10.1002/chem.200800239, and 10.1002/mrc.932).
- Although the existing shortcomings in grammar do not hinder understanding, it is advisable to eliminate them.
Author Response
Dear reviewer,
Thank you very much for taking your precious time to revise my article. Your comments are very helpful to me, and the following are modifications based on your suggestions. Thank you again for your help.
Best wishes.
Please see the attachment.

Reviewer 2 Report
In the paper the synthesis and characterization of three different poly(arylene ether)s with POSS in the main chain is reported. These system are quite interesting and the presence of POSS produces improvement in the properties of the materials, contributing to a lower dielectric constant and a better thermal and chemical resistance. The paper is worth to be published in Polymers, after some minor corrections.
- Abstract: the first sentences "A series of crosslinked poly(arylene ether)s with POSS in the main chain are reported here. In this study, A series of fluorinated terminated poly(arylene ether)s were firstly ..." need syntax correction.
- Lines 69: "Chemical Co. Ltd.. Used after recrystallization" --> "Chemical Co. Ltd. and used after recrystallization".
- The solvent used for the NMR analysis should be indicated in the Experimental part.
- Line 100: "and wash" --> "and washed".
- Figure 5: check the aspect ratio.
- Figure 6: The inflection point of Tg is not visible. Enlarge the vertical scale.
- Lines 185 and 186: "WARD" --> "WAXRD".
Author Response

(The authors gave the same response as above.)

Round 2
Reviewer 1 Report
It is not clear to me why some of the authors' responses (Points 1, 3 and 4) were not included in the manuscript. I think this might be useful for readers. However, since this information is not critical to understanding the results of the article, I leave that decision to the authors.
Author Response
Dear reviewer,
Thank you very much for your valuable suggestions . Your comments are very helpful to me. I have revised the manuscript and included important information (points 1, 3, and 4) in the manuscript. Thank you again for your help.
Best wishes.